# Stress-Induced Phosphaturia in Weaned Piglets

**DOI:** 10.3390/ani10122220

**Published:** 2020-11-26

**Authors:** Malgorzata Habich, Bartosz Pawlinski, Maria Sady, Katarzyna Siewruk, Piotr Zielenkiewicz, Zdzislaw Gajewski, Pawel Szczesny

**Affiliations:** 1Department of Bioinformatics, Institute of Biochemistry and Biophysics PAS, ul. Pawinskiego 5A, 02-106 Warsaw, Poland; malgorzata.habich@ibb.waw.pl (M.H.); piotr@ibb.waw.pl (P.Z.); 2Department of Large Animals with Clinic, Warsaw University of Life Sciences, Nowoursynowska 100, 02-797 Warsaw, Poland; pawlinski_b@op.pl (B.P.); m.sady92@gmail.com (M.S.); siewruk.katarzyna@gmail.com (K.S.)

**Keywords:** weaning period, hypophosphatemia, tissue hypoxia, biochemical parameters

## Abstract

**Simple Summary:**

The weaning period is a critical period in piglets’ lives. Multiple elements, including diet change, social stress, handling, and change of physical environment, contribute to enormous stress that has health implications. This period has been studied extensively in the past, but some gaps in our knowledge remain. We attempted to fill them by biochemical characterizations of the changes in blood and urine before and after the weaning. The major finding is the observation of the release of phosphate in the urine in the apparent absence of a factor other than weaning. This release is followed by the drop of the phosphate in the blood. Additionally, we observed a population-level Bohr effect, suggesting a decrease in oxygen levels in the tissues. These results point to the development of systemic hypophosphatemia, even though modern diets used in pig breeding have an excess of phosphorus typically. This study sheds new light on the weaning period and will help researchers and veterinary practices in improving design studies and treatments around that time.

**Abstract:**

The weaning period in piglets draws significant attention from researchers, veterinarians, and breeders. A substantial change in diet accompanied by enormous stress has health and welfare implications (abnormal feeding intake, infections, umbilical lesions, etc.). While the parameters like optimal age or weight for the weaning have been studied extensively, relatively less attention has been devoted to the study of stress effects in the piglets’ biochemistry. As one of the effects of stress is hyperventilation, a gasometric analysis supported by measurements of hypoxia biomarkers was conducted. Piglets blood and urine, one day and seven days before and one day and seven days after the weaning, were tested. There was no evidence of hyperventilation, but phosphaturia and hypophosphatemia were observed one and seven days postweaning, respectively. A statistical analysis across the population also pointed to minor tissue hypoxia. Our work contributes to an understanding of biochemical dynamics and helps in the interpretation of physiological changes observed in piglets in this critical period.

## 1. Introduction

Piglets encounter a number of challenges during weaning, including a different food source; social stress (separation from the sow and commingling with pigs from other litters); transportation and handling stress; a different physical environment (room, building, farm, water supply, etc.); increased exposure to pathogens; dietary or environmental antigens; and, often, additional medical treatments such as castration. Insufficient environmental conditions and poor management practices can lead to increased mortality, morbidity, and retarded growth [1].

The change in life corresponds with changes within pigs’ physiology and behavior: reduced food intake during the first week after weaning, small intestine modifications (villous atrophy and crypt elongation), and reduced brush-border digestive enzyme activity. Those changes can contribute to common postweaning complications, such as diarrhea. Studies also report an increase in the corticotropin-releasing factor (CRF) and cortisol levels, as well as the upregulation of proinflammatory cytokines [2].

A human stress response is a well-described phenomenon and exhibits many similarities to the changes we observe in pigs [3]. The very characteristic reaction for humans to stress is an increased breathing rate. The immediate response to hyperventilation is the decline in serum CO_2_ and the rise of O_2_, resulting in respiratory alkalosis. After a few hours, compensatory mechanisms balance the serum pH by renal acid retention, causing a drop in plasma HCO_3_^−^. Although it is known that pigs hyperventilate in response to heat stress, it is not clear if the response is the same in the case of weaning. The only biochemical change that could be attributed to hyperventilation so far is an observation of a rise in cortisol level during the weaning phase [4].

In this study, we examined a number of biochemical markers of possible stresses associated with weaning in piglets, with emphasis on those likely to be associated with hyperventilation. We found that changes in serum O_2_ and HCO_3_^−^ were negligible, which suggests that hyperventilation is not a concern during the weaning period. However, we found phosphaturia and evidence of tissue hypoxia. Given a trend to keep the phosphorus level in feeding diets very close to the requirements, this data can be crucial during the first period of food transition.

## 2. Materials and Methods

### 2.1. Ethical Approval

The study was approved by the Third Local Ethical Committee on Animal Testing in Warsaw, Poland (permit number: 20/2015 from 23 April 2015) on behalf of the National Ethical Committees on Animal Testing. Each animal was used in the experiment only once to reduce stress.

### 2.2. Animals and Experimental Design

The study was conducted on 120 healthy (female ♀, male ♂), randomly selected Polish Large White (PLW) breed piglets at the age of up to six weeks. To avoid confounding factors stemming from a particular approach to animal handling, we used two farms in two distinct areas of the country. One hundred and twenty healthy piglets (sixty from each farm) at the age until six weeks were divided into four groups: I (*n =* 14) 7 days before weaning, II (*n =* 16) at the day of weaning, III (*n =* 46) one day after weaning, and IV (*n =* 44) 7 days after weaning (Table 1). Piglets during the experiment were fed using commercial solid “starter” feds available at all times in feeders (De Heus, Leczyca, Poland). The sows were fed three times a day. Tap water was allowed ad libitum. Environmental conditions, such as temperature and humidity, were adjusted automatically. The temperature for lactating sows was between 18 and 20 °C, and piglets were heated by radiant heaters (26–30 °C). During the entire study, the sows and piglets were clinically healthy.

### 2.3. Anesthesia Procedure and Blood Biochemical and Gasometric Analysis

To minimize the handling stress, animals were premedicated with an intramuscular injection of azaperone (Stresnil, 3 mg/kg body weight (b.wt.), Janssen Pharmaceutica, Tumhoutseweg, Belgium). Blood samples were collected after sedation (a few minutes after administration of Stresnil) from the brachial vein collected into vacutainer tubes (1-mL gasometric tube and 3-mL biochemical tubes, Moneovette 1ml LH, Sarstedt, Numbrecht, Germany). Oximetric parameters were assessed in whole blood with a critical points analyzer RAPIDPoint 500 (Siemens, Erlangen, Germany), and biochemistry parameters were assessed with biochemical analyzer BS-120 (Midray DP-20, Transduce 3.5 MHz 35C50EB, Shenzhen, China). During the experiment, the following parameters were marked: anion gap (mmol/L), FMetHb (%), urea (mg/dL), pCO2 (mmHg), pH, pO1 (mmHg), BE(B) (mmol/L), Cl (mmol/L), FCOHb (%), FHHb (%), FO2Hb (%), HCO3-act (mmol/L), Hct (%), K (mmol/L), Na (mmol/L), phosphorus (P) (mg/dL), Ca (mmol/L), glucose (mg/dL), Mg (mg/dL), sO2 (%), and tHb (g/dL). Urine samples were collected from the bladder under ultrasound control into a sterile tube (Novama, Lodz, Poland). The following parameters were marked: uric acid (mg/dL), P (mg/dL), Ca (mmol/L), glucose (mg/dL), creatine (mg/dL), and Mg (mg/dL).

### 2.4. Statistical Analysis

All analyses were done using Python 3.5. Statistical significance was determined with the two-sample Kolmogorov-Smirnov test from the SciPy library.

## 3. Results

All measurements were conducted according to the methods described in the Materials and Methods section. Data were obtained from two different herds to limit selection bias. Serum and urine test results were comparable with biochemical reference values from previously published data [4,5] (see Figure 1). For example, Yu and coworkers reported glucose on the weaning date of 84.9 ± 17.8 mg/dL, while, in the study of Perri and coworkers, one day before weaning glucose was 118 mg/dL. In our experiments, one day before weaning, the blood glucose was 122.5 ± 27.4 mg/dL. The same similarities were observed in the arterial blood gas test data [6,7]. For example, the bicarbonate concentration reported by Harris and coworkers was 28.4 ± 0.5 mEq/L and, by Hannon and coworkers, was 31.6 mEq/L, while, in our study, it was 26.2 ± 2.9 mEq/L. Minor differences in the observed parameters could be attributed to differences in age, feeding pattern, or food composition. Unfortunately, it was not possible to standardize the urine test results with creatine due to the lack of ability to weigh piglets during the experiment.

The mean concentration (ng/mL) of ochratoxin in the serum in the groups (−7, −1, +1, and +7) was 1.38, 1.01, 0.3, and 0.05, which was insufficient to induce nephropathy or other health problems related to mycotoxin during the experiment [8].

Since the goal of the study was to assess the stress biomarkers related to respiration, we carefully analyzed the arterial blood gas test data. If the herd suffered from hyperventilation, we would suspect a rise in pH, pO_2_, FO_2_Hb, and sO_2_ and decline in pCO_2_ and HCO_3_^−^. Although we observed some statistically significant differences between the groups, the magnitude and direction of changes did not fit into the hyperventilation pattern. There was a substantial drop in pO_2_ (94 mmol/L before weaning vs. 78 mmol/L after weaning, *p*-value: 9 × 10^−5^) instead of the raise expected during hyperventilation. Additionally, the pH slightly dropped, whereas we would expect posthyperventilation respiratory alkalosis. To understand the apparent drop in partial oxygen pressure, we analyzed the oxygen dissociation curves between two populations, looking for evidence of the Bohr effect. The postweaning group exhibited a minor but significant (pO_2_/sO_2_ values compared between two populations yielded a difference of 0.3 with a *p*-value of 0.003) shift to the right (Figure 2).

This minor tissue hypoxia could be explained by phosphaturia (an increase of urine phosphate levels) observed one day after weaning, followed by hypophosphatemia (decrease of blood phosphate levels) on day seven, as hypophosphatemia is known to induce tissue hypoxia [9]. Our assumption on the presence of phosphaturia was based on comparisons between the postweaning and preweaning groups (see Table 2 below). It has to be noted that a substantial increase in urinal phosphate in group +1 as opposed to group −1 was assessed as insignificant due to the large variation in the former group. To verify the hypothesis on tissue hypoxia, we divided the animals into two groups not in respect to weaning but in respect to blood phosphate levels. We assumed that the 25th quantile of the phosphate level would define a group with lower phosphate, which would presumably have stronger tissue hypoxia. The repeated analysis of differences of the oxygen dissociation curves between “hypophosphatemic” and “normophosphatemic” groups resulted in a larger effect size (0.4) with a better *p*-value (7 × 10^−5^).

Full characterization of the two comparisons described above: hypophosphatemic vs. normophosphatemic and postweaning vs. preweaning is shown in Table 2. The pH, sO_2_, and pO_2_ consistently and significantly dropped in the two analyzed comparisons, further supporting the tissue hypoxia hypothesis. Interestingly, except for the blood magnesium and blood anion gap, the magnitude of the changes in the comparison between hypophosphatemic and normophosphatemic animals was smaller (and associated *p*-values larger) than in comparison between postweaning and preweaning.

## 4. Discussion

The chosen parameters were standard and sufficient to observe hyperventilation [10,11]. Yu and coworkers reported that cortisol was significant after weaning in piglets, which was an indication that weaning was indeed received as a stressful situation by the piglets. The dynamic of the change of the concentration of cortisol in the blood was much faster than the organisms’ reaction to the changed acid-base balance after respiratory alkalosis. Full recovery from a severe episode required two to three days for completion [9]. If hyperventilation had occurred, we would have seen the change in the blood gas panel at least in the +1-day group.

Overall, we concluded that no evidence of hyperventilation in the experiment was found. While some of the observed changes in the biomarkers were statistically significant, not all of them fit the hyperventilation pattern. For example, the blood pH decreased slightly in the postweaning group, while we would expect a sharp increase. The temperature was controlled, minimizing the possibility of heat stress.

Previous studies showed [12,13] that injury can induce hypophosphatemia due to phosphaturia in pigs, but there was a lack of evidence that other stressors can result in similar reactions, and the effects can be observed for a prolonged time. There was evidence that an increased secretion of cortisol via a 12-h-long adrenocorticotropic hormone infusion in cows disrupted the homeostasis of inorganic phosphate and resulted in hypophosphatemia [14]. Yu and coworkers reported that cortisol was significant after weaning in piglets, which was an indication that weaning was indeed received as a stressful situation by piglets. The dynamic of the change of the concentration of cortisol in the blood was typically much faster than the organisms’ reaction to the changed acid-base balance after respiratory alkalosis. However, in the absence of aggressive behaviors in studied herds (no injuries were noted), a plausible hypothesis is that elevated cortisol without associated hyperventilation could be a reason for the observed disturbances of phosphate metabolism. No other stress-response-associated mechanism is known for causing hypophosphatemia.

Concluding, our findings showed a significant problem of hypophosphatemia in the population of postweaning pigs, which can be explained by preceding phosphaturia. Severe phosphaturia was observed even seven days after weaning, but the extent of the effect is unknown and should be further investigated.

Minor tissue hypoxia in postweaning piglets was identified using a population-level analysis of the Bohr effect. The effect increased if, instead of a post- and preweaning comparison, we assessed the hypophosphatemic and normophosphatemic animals. The oxygen levels were lower in the hypophosphatemic and postweaning groups consistently and with statistical significance.

Postweaning piglets are prone to infections because of the change in environment and increased exposure to pathogens [15]. One of the most common symptoms of general infection is an increased sensitivity to fungus [16]. Ochratoxin A is formed by different *Aspergillus* and *Penicillium* species, and it is one of the most abundant food-contaminating mycotoxins and induces a strong nephrotoxic effect [17]. As an additional control against the possibility that observed changes after the weaning period are a result of a nephrotoxic effect rather than weaning itself, we assessed the concentration of popular mycotoxin in the serum. As the levels of ochratoxin were minimal, it is likely that the infection played no role in phosphaturia and hypophosphatemia.

The dietary phosphorus (P) supply has attained an increasing interest in farm animal nutrition during the last decades due to environmental concern and to limited global phosphate resources [18,19,20]. Phosphorus and other micronutrients are vital elements for all living organisms. They play a fundamental role in growth processes, the formation and stability of bones, and maintenance of physiological cell metabolism. Novel findings in pigs also provide increasing evidence for a link between dietary P and the immune system [21]. Phosphorus is an irreplaceable component of life and, thus, widely used in all agricultural production systems. However, in pig husbandry, the dietary P levels often exceed animal and age-specific requirements [22]. Gerlinger et al. [23] concluded during their research that the animals on a high Ca and P diet exhibited reductions in growth performance and health status. An increase in the dietary P content had no positive effect on the bones or overall pig performances [23]. The economic and ecological aspects are convincing for the minimum phosphorus level in pig nutrition. Pigs are considered to be major excretors of P from agricultural systems, which leads to a severe environmental burden. In order to balance the economic and environmental sustainability related to the uneven density of animal production, novel approaches of P management aim to slow down the rate of emissions through transformative or incremental system-wide processes [23]. We should strive for feeding with a balanced phosphorus feed, maintaining the proper development of piglets. During our research, hypophosphatemia did not seem to harm the development of piglets and their lives. However, a seven-day span might not be enough to observe adverse effects, like a smaller weight gain. The piglets were on a commercial “starter” fed available at all times; therefore, the effects on phosphate homeostasis could not be attributed to dietary insufficiencies.

## 5. Conclusions

Significant phosphaturia in postweaning piglets translated into hypophosphatemia observed seven days later. Minor tissue hypoxia was observed alongside. We ruled out the nephrotoxic effects of a fungus infection as the cause; therefore, the observed changes should be considered a part of the stress response related to weaning itself. We found no evidence that the observed biochemical changes can be attributed to hyperventilation. Therefore, the underlying cause remains to be discovered. However, in the absence of dietary insufficiencies of phosphorus and a lack of support for such a thesis in longitudinal studies, we conclude that observed hypophosphatemia might not require correction in the course of piglets’ development. Instead, further research should focus on revealing the mechanism of the weaning-induced stress.

## Figures and Tables

**Figure 1 animals-10-02220-f001:**
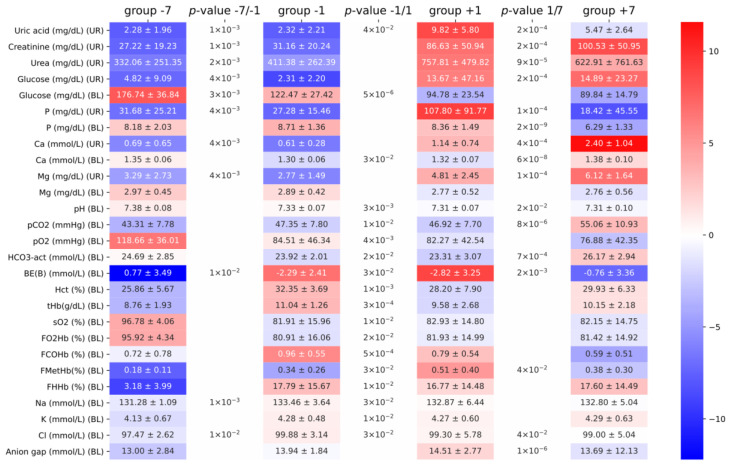
Summary of the observed differences of blood (BL) and urine (UR) biochemical parameters. Groups −7, −1, +1, and +7 indicate seven and one days before weaning and one and seven days postweaning, respectively. Numbers between cells denote the significance of pairwise comparisons between two groups, and only significant (<0.05) values are shown. Colors show a row-wise logarithm of the proportion of the mean value of the parameter within a group to the mean value of all measurements (intuition: how far a group is from the mean of all pigs in a logarithmic scale).

**Figure 2 animals-10-02220-f002:**
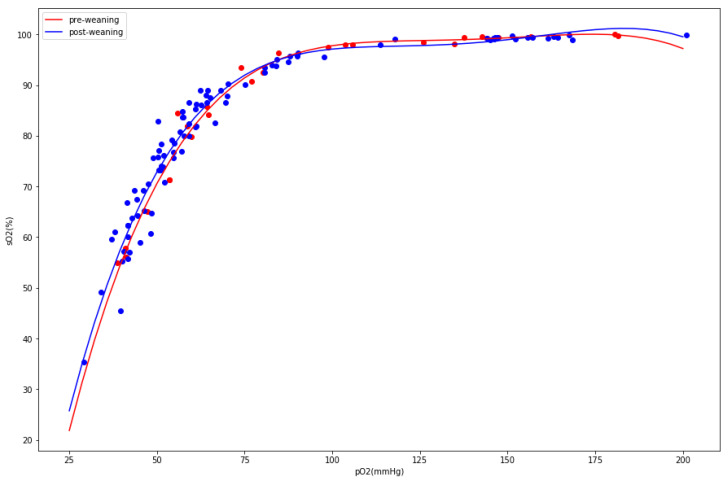
Population-level oxygen dissociation curves for pre-weaning (red) and post-weaning (blue). Individual measurements marked with dots.

**Table 1 animals-10-02220-t001:** The division into groups according to age.

Group	Time to Weaning	N (Numbers)
Group I	Seven days before	14 (♀ = 0, ♂ = 14)
Group II	At the day	16 (♀ = 3, ♂ = 13)
Group III	One day after	46 (♀ = 10, ♂ = 36)
Group IV	Seven days after	44 (♀ = 4, ♂ = 40)

**Table 2 animals-10-02220-t002:** Statistical analysis of differences in the biochemical parameters for two comparisons: (1) between the hypophosphatemic and normophosphatemic groups and (2) between the postweaning and preweaning groups. BL and UR correspond to blood and urine, respectively, as in Figure 1. NA (not available) in the case of blood phosphate in the comparison between hypophosphatemic and normophosphatemic denotes a case where this parameter was selected to differentiate between the groups. N.S. indicates a nonsignificant change (*p*-value > 0.05). P: phosphorus. Arrows indicate the direction of the change.

Biomarker	Hypophosphatemic vs. Normophosphatemic		Postweaning vs. Preweaning	
	*p*-Value	Change	*p*-Value	Change
Uric acid (mg/dL) (UR)	0.02	↑	6.39 × 10^−8^	↑
Creatinine (mg/dL) (UR)	0.005	↑	8.23 × 10^−8^	↑
Urea (mg/dL) (UR)	0.01	↑	0.0001	↑
Glucose (mg/dL) (UR)	0.002	↓	0.002	↑
Glucose (mg/dL) (BL)	0.002	↓	2.44 × 10^−15^	↓
P (mg/dL) (UR)	0.02	↑	3.48 × 10^−6^	↑
P (mg/dL) (BL)	NA	NA	2.14 × 10^−5^	↓
Ca (mmol/L) (UR)	0.02	↑	0.0001	↑
Ca (mmol/L) (BL)	0.003	↑	0.15	N.S.
Mg (mg/dL) (UR)	0.03	↑	3.73 × 10^−5^	↑
Mg (mg/dL) (BL)	7.53 × 10^−6^	↓	0.29	N.S.
pH (BL)	0.02	↓	5.11 × 10^−7^	↓
pCO2 (mmHg) (BL)	0.02	↑	0.003	↑
pO2 (mmHg) (BL)	0.0002	↓	9.62 × 10^−5^	↓
HCO3-act (mmol/L) (BL)	0.19	N.S.	0.003	↑
BE(B) (mmol/L) (BL)	0.26	N.S.	5.48 × 10^−5^	↓
Hct (%) (BL)	0.14	N.S.	0.001	↓
tHb(g/dL) (BL)	0.14	N.S.	0.0004	↓
sO2 (%) (BL)	0.0001	↓	4.97 × 10^−5^	↓
FO2Hb (%) (BL)	0.0002	↓	2.77 × 10^−5^	↓
FCOHb (%) (BL)	0.44	N.S.	6.03 × 10^−5^	↓
FMetHb (%) (BL)	0.9	N.S.	0.003	↑
FHHb (%) (BL)	0.008	↑	0.002	↑
Na (mmol/L) (BL)	0.66	N.S.	0.15	N.S.
K (mmol/L) (BL)	0.06	N.S.	0.04	↑
Cl (mmol/L) (BL)	0.95	N.S.	0.15	N.S.
>Anion gap (mmol/L) (BL)	>0.0003	>↓	>0.001	>↑

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
