# Peer review of "Stress-Induced Phosphaturia in Weaned Piglets"

_animals, 2020, doi:10.3390/ani10122220_

Round 1

Reviewer 1 Report

Just some minor editing required.

  1. Simplify to “weaning is a critical period …..
  2. …typically have an excess of phosphorus
  3. “optimization of phosphate usage” I suggest it would be clearer to state “trend to feeding diets with phosphate levels very close to requirements” or “without excess phosphate”

I don’t think “optimization” is the correct term here

  1. “change in environment”
  2. 174. Not sure what is meant by “controlled the concentration of popular mycotoxin in serum?

Should this be simply that we “assessed the concentration of ochratoxin in the serum?” to determine if this was an important factor in the response

Author Response

Just some minor editing required.
12. Simplify to “weaning is a critical period …..
Line 12: replaced “moment” with “period.”

13. …typically have an excess of phosphorus
Line 21: replaced “phosphate” with “phosphorus.”

14. “optimization of phosphate usage” I suggest it would be clearer to state “trend to
feeding diets with phosphate levels very close to requirements” or “without excess
phosphate”I don’t think “optimization” is the correct term hereFully agreed. The sentence (Lines 63-64) states now “Given a trend to keep phosphorus level
in feeding diets very close to requirements (…).”.

169. “change in environment”
Line 214: replaced „of the” with “in.”

170. 174. Not sure what is meant by “controlled the concentration of popular
mycotoxin in serum?
Should this be simply that we “assessed the concentration of ochratoxin in the serum?” to determine if this was an important factor in the response.
Yes indeed, that was the intention. Line 219, replaced “controlled” with “assessed.”

Reviewer 2 Report

This paper reports a study of biochemical changes in piglets associated with weaning. When presented without preconceptions it contains some useful information, especially relating to P in urine and blood. The title reflects this. The text, however, is overloaded with the presumption that weaning might cause hyperventilation and thereby respiratory alkalosis, based on the premise that hyperventilation is a response to fear, which  may or may not occur as one of the many possible stresses of weaning. The emphasis on hyperventilation is weakened further by the fact that you did not measure respiration rate.

The P results are interesting and worthy of publication. The most conspicuous omission from your text is mention of the extremely large variation in [P] urine at W+1, which rendered the day O to day 1 difference insignificant.  Your current Fig 2 illustrates nothing of physiological significance. However you briefly state (l139-142) that you analysed the low [P]blood (or high [P] urine?) group separately and showed a larger difference in oxygen saturation curves. This analysis shpud be developed and would make for a more interesting Fig 2.

I believe this paper contains information worthy of publication.. However discussion of (fear-induced?) hyperventilation should be minimised. More discussion of (non-fear?) reasons why only some pigs showed large changes in [P] could be valuable. It would be interesting to know whether the low[P] group showed  large shifts in other measured parameters.

Author Response

We thank the reviewer for the insightful comments. Indeed respiration rate wasn’t measured,
but we looked for biochemical markers of hyperventilation and we found no evidence of it, which is clearly stated in the conclusions. That said, we agree that we discuss hyperventilation
too much, given lack of evidence. Therefore, the whole paragraph on the hyperventilation was
removed from the discussion, as it indeed did not contribute substantially to the conclusions
of the paper (Line 170). This is accompanied by a slight restructuring of the discussion and
pointing to the cortisol, as the only plausible hypothesis left that explains hypophosphatemia
observed post-weaning (Lines 198-206).
We added a note on the insignificant difference of urine phosphate between day 0 and day +1
(Lines:141-145) and added the table of comparisons (line 151): hypophosphatemic-
normophosphatemic and postweaning-preweaning. That clarifies the apparent omission of the
discussion on urine phosphate in respect to Figure 1 (and explains why we considered an
evidence of phosphaturia robust nevertheless). Additionally, this table addresses the next
comment on the development of the conclusions stemming from Fig. 2. Fig. 2 developed for
the comparison between hypophosphatemic and normophosphatemic animals looks quite
similar to the postweaning vs preweaning comparison. It did not make sense in our view to
produce two almost identical figures. Therefore, we added a table and commentary that
highlights the consistency between oxygen-related biomarkers changes in both
aforementioned comparison (lines 160-165). This is accompanied by a summary in the
discussion (Lines: 211-214).

Round 2

Reviewer 2 Report

The key message from this revised MS, containing the new Table 1 is that some (not all) piglets exhibited phosphaturia after weaning, and that this was linked to hypoxia. Since this result was the opposite of that which would occur as a result of hyperventilation, you must conclude that there was absolutely no suggestion that this occurred as a consequence of weaning stress. The fact that some piglets exhibited phosphaturia is interesting but cannot be explained by the data.  There is a case for further study incorporating behavioural measurements and perhaps two more or less potentially stressful weaning procedures

L69-73. Amended final paragraph of introduction  must be rewritten, not least because it refers to results. I suggest ‘in this study we examine a number of biochemical markers of possible stresses associated with weaning in piglets, with emphasis on those likely to be associated with hyperventilation.

L163. It would make more sense to refer to the pre and post weaning groups in chronological order as Group 1 and 2 respectively.

Table 1 is valuable but the caption and column headings need to be improved.

Caption: Significance (P values) of the effects of hypophosphataemia on other biochemical markers assessed by comparisons between i) preweaning and day1 postweaning values for all pigs, ii) hypo v. normal blood phosphorus (P) values on Day 1 post weaning, 

Headings to columns for Table 1

____________________________________________________

Significance of effects of hypophosphataemia (hypoP v. normal P)

            Biomarker            Within D+1                D-1 v. D+1

                                   P value  Change        P value   Change

____________________________________________________

            Uric acid              0.02       +            <0.001         +

            HCO3                 n 0.19     N.S.           0.003       N.S.

Author Response

The key message from this revised MS, containing the new Table 1 is that some (not all) piglets exhibited phosphaturia after weaning, and that this was linked to hypoxia. Since this result was the opposite of that which would occur as a result of hyperventilation, you must conclude that there was absolutely no suggestion that this occurred as a consequence of weaning stress. The fact that some piglets exhibited phosphaturia is interesting but cannot be explained by the data.  There is a case for further study incorporating behavioural measurements and perhaps two more or less potentially stressful weaning procedures

L69-73. Amended final paragraph of introduction  must be rewritten, not least because it refers to results. I suggest ‘in this study we examine a number of biochemical markers of possible stresses associated with weaning in piglets, with emphasis on those likely to be associated with hyperventilation.

Lines 70-72 – the paragraph amended as suggested.denotes case

L163. It would make more sense to refer to the pre and post weaning groups in chronological order as Group 1 and 2 respectively.

Line 162 – groups listed in chronological order.

Table 1 is valuable but the caption and column headings need to be improved.

Caption: Significance (P values) of the effects of hypophosphataemia on other biochemical markers assessed by comparisons between i) preweaning and day1 postweaning values for all pigs, ii) hypo v. normal blood phosphorus (P) values on Day 1 post weaning, 

Headings to columns for Table 1

____________________________________________________

Significance of effects of hypophosphataemia (hypoP v. normal P)

            Biomarker            Within D+1                D-1 v. D+1

                                   P value  Change        P value   Change

____________________________________________________

            Uric acid              0.02       +            <0.001         +

            HCO3                 n 0.19     N.S.           0.003       N.S.

We thank the reviewer for that comment because that indicates that the structure of the table is misleading. The table contains two comparisons: one is between hypo- and normophosphatemic groups, and the other is between postweaning and preweaning. We didn’t additionally use third dimension of days (too small size in compared groups). The headers were updated in analogous way to reviewer’s suggestion but reflecting the comparison presented in the table. We included the N.S. notation for non-significant results and the table main caption was improved to state: “Statistical analysis of differences in biochemical parameters for two comparisons 1.) between hypophosphatemic and normophosphatemic groups and 2.) between post-weaning and pre-weaning groups.”

This manuscript is a resubmission of an earlier submission. The following is a list of the peer review reports and author responses from that submission.

Round 1

Reviewer 1 Report

General comments

Simple summary

The authors state that diet change is an important factor contributing to the stress at weaning, and discuss phosphate use in diets, however, diet is then ignored in the research.   The simple summary needs to provide more information as to the importance of the findings, than just to state that they are important.

Abstract

Limit the use of “we” in the abstract. It is not clear to me what the actual objectives of the study are. I suggest the focus should be more on parameters indicative of stress, such as evidence of hyperventilation. Why were the specific parameters chosen? (other than being easily analyzed). What is the hypothesis?

Introduction

The focus is on stress response. It is not clear to me if the final paragraph/sentences are referring to the current work, or to previous work by these authors. The introduction does not lead into a defined hypothesis.

Animals and Experimental design

It is very unclear why two different farms were used, how many pigs per farm? Feeding 2x per day implies that feed was not available at all times, - is this correct?

Results

When the results and discussion section are presented separately, this section should only contain results (ie. lines 107 to 115 belong in the discussion)

Details

Line 6. Is this really “with” clinic?

  1. 11. Weaning is a critical period in a piglets life.
  2. 12. Since you are focusing on stress, - I suggest you mention other factors besides diet changes that can contribute even more to stress
  3. 14. I believe you are contributing data, - but I don’t feel this is a “comprehensive biochemical assessment”
  4. 17. How was reduced O2 levels in tissue found? You only measured blood and urine, not tissues
  5. 19. I don’t think this is important. Suggest the first two lines are reworded and focus on the pig
  6. 23. Replace “biochemical trajectory” with biochemistry
  7. 25. , day and seven days before and …….. please reword this entire sentence
  8. 26. Replace with “there was no evidence of hyperventilation, but phosphaturia and hypophosphatemia was observed one and seven days post weaning, respectively.
  9. 34. Reword. Ie. Piglets encounter a number a challenges during weaning, including……..
  10. 41 – 43. Provide references
  11. 45. Increase in serum corticotropin-releasing factor
  12. 51. What is the loss of biocarbonate referring to here?
  13. 52. In response to heat stress
  14. 53. heard reaction?
  15. 68. What is meant by “according to the production cycle”
  16. 75. What was azaperone used? How soon was blood collected after it was used? How do you think this might affect stress?
  17. 87, the experimental design, (ie. were the different farms incorporated into the model) needs to be included.
  18. 89. Would it not be helpful to standardize urine values using creatinine?
  19. 112. Why was ochratoxin A chosen as a mycotoxin of interest. Other mycotoxins (ie. DON) are more commonly found in feeds throughout Eastern Europe
  20. 166. The discussion on dietary phosphorus supply is interesting, especially since the main result was phosphaturia. Thus it is imperative that phosphate levels (calculated available) of the diets used in these studies are added to the paper.
  21. 189. What is meant by “in the absence of dietary insufficiencies of P”, - are you referring to your study specifically or is this a general statement. is the overall conclusion that the observed hypophosphatemja is not important? Stress is not evident at weaning? Without a hypothesis, - it is difficult to determine a conclusion.

Reviewer 2 Report

The hypothesis is that weaning would cause hyperventilation induced by 'stress'.  There are many known stressors associated with weaning that can have multiple effects. Hyperventilation could be an indicator of stress induced by fear.  Moderate hyperventilation during heat stress is an adaptation to increase evaporative heat loss from the respiratory tract.  In an experiment designed, I think, to distinguish between fear and heat stress, there were no recorded measurements of either respiration rate or air temperature.

Samples were taken of blood and urine. Fig 1 does not distinguish between the two. Values for (e.g.) P appear twice but the Fig does not say which is which. This makes it impossible to interpret the changes in P, which appears to have chosen retrospectively as the key finding of this study.

The absence of any shift in blood pH is an important observation but goes without comment. 

The inclusion of discussion (in the results section l107-112) is irrelevant.